# Effect of High-Temperature Deformation Twinning on the Work Hardening Behavior of Fe-38Mn Alloy during Hot Shear-Compression Deformation

**DOI:** 10.3390/ma17153641

**Published:** 2024-07-24

**Authors:** Deli Sang, Xiaoli Xin, Zikang Zhai, Ruidong Fu, Yijun Li, Lei Jing

**Affiliations:** 1College of Materials Science and Engineering, Shijiazhuang Tiedao University, Shijiazhuang 050047, China; 2State Key Laboratory of Mechanical Behavior, System Safety of Traffic Engineering Structures, Shijiazhuang Tiedao University, Shijiazhuang 050043, China; 3State Key Laboratory of Metastable Materials Science and Technology, Yanshan University, Qinhuangdao 066004, China; 4Northwest Institute for Nonferrous Metal Research, Xi’an 710016, China

**Keywords:** severe plastic deformation, high temperature, deformation twinning, work hardening behavior

## Abstract

The effect of high-temperature deformation twinning on the work hardening behaviors of Fe-38Mn alloy during hot shear-compression deformation was investigated. The discovery of micro-shear bands and deformation twinning is significant for continuous work hardening, and this represents an important step toward gaining a complete understanding of the effect of deformation twinning on work hardening behaviors. Deformation twinning is widely acknowledged to accommodate plastic strain under cold deformation, even under severe plastic deformation. At present, the equivalent stress vs. strain curves for hot shear-compression deformation of Fe-38Mn alloy exhibit the characteristics of continuous work hardening. In addition, continuous work hardening is classified into five stages when considering high-temperature deformation twinning.

## 1. Introduction

Metals generally undergo strain or work hardening during plastic deformation, meaning that the flow stress of the metals continually varies with plastic strain [1]. The work hardening behavior of hot deformation is the result of competition between work hardening caused by the form of dislocation and dynamic microstructural softening caused by the annihilation of dislocations taking place either through dynamic recovery (DRV) or dynamic recrystallization (DRX) [2,3,4,5]. In addition, thermal softening due to the strain rate leads to strain localization. Therefore, in general, the theoretical work hardening rate curve of thermal deformation is divided into different stages, which is related to but not limited to dislocations, dynamic microstructural softening and thermal softening [6,7,8,9].

Twinning usually occurs to coordinate deformation when it would be difficult for dislocation slip to proceed. For a face-centered cubic (FCC) crystal with low stacking fault energy (SFE), twinning is a common occurrence through layer-by-layer displacement of a/6 <112> Shockley partial dislocations on consecutive {111} planes via the pole mechanism, stair-rod cross-slip mechanism or three-layer mechanism [10,11,12,13]. Twinning contributes significantly to work hardening behavior in low-SFE metals [14,15,16,17]. The twin boundary can not only effectively hinder dislocation motion but also accommodate the dislocation through interaction with dislocation to adjust the plastic strain, especially in the deformation twinning during room-temperature or low-temperature deformation [18,19,20]. In order to study the work hardening behavior under a large strain in low-SFE FCC alloys, Asgari et al. [21] developed and presented a physical description of the deformation twinning responsible for the various work hardening stages observed through room-temperature compression of a-brass and MP35N. Then, the twin volume fraction, twinning, twin–dislocation interactions and twin–twin interactions became the central topics in the field of work hardening behavior. Furthermore, Vinogradov et al. [22] presented a model of strain hardening in materials with twinning-mediated plasticity. So, here is the question: How does deformation twinning affect the work hardening behavior during hot deformation? As a starting point, the key role of deformation twinning in dynamic microstructural evolution was confirmed during hot deformation in our previous studies [23,24,25].

In the present study, Fe-38Mn alloy was taken as a model material for its low SFE and single-phase austenite. Hot shear-compression deformation (HSCD) was employed to realize a large strain and high strain rate deformation, to introduce the occurrence of high-temperature deformation twins, which verified the feasibility of the physical simulation of combined hot deformation in transient conditions [26,27,28]. The effect of high-temperature deformation twinning on the work hardening behavior was investigated in detail.

## 2. Experimental Section

The chemical components of the presented Fe-38Mn alloy are (in *wt* pct) 0.063 C, 38.38 Mn, 0.003 S, 0.011 P, 0.030 Si and balanced Fe. The hot shear-compression deformation (HSCD) of Fe-38Mn alloy was performed on a Gleeble 3500 thermo-mechanical simulator (Dynamic Systems Inc. (DSI), Poestenkill, NY, USA) at axial strain rates of 0.01 s^−1^, 1 s^−1^ and 10 s^−1^ while the deformation temperature ranged from 900 °C to 1100 °C. With a heating rate of 20 °C/s, the specimens were quickly heated to the deformation temperature and held for 30 s. After that, they were deformed with a total axial compression displacement of 4.5 mm (equivalent strain was 1.4 [26]) and then quenched in water immediately. A schematic representation of a shear-compression specimen (SCS) is shown in Figure 1. Three samples were tested under each parameter, and the one with the best deformation was taken as a reference. The microstructure was examined by optical microscopy (OM, observed after mechanical polishing and corrosion with 5% nitrate alcohol), scanning electron microscopy (SEM) using the electron backscattered diffraction technique (EBSD, observed by electrolytic polishing solution with perchloric acid: glycerol: perchloric acid 3:1:17) and transmission electron microscopy (TEM, observed after double spraying with 15% perchloric acid alcohol electrolysis).

## 3. Results

### 3.1. Flow Behavior

The conversion relationship between the equivalent strain–equivalent stress curve and axial displacements and loading forces is referred to in references [26,27] and will not be detailed here. The equivalent stress vs. equivalent strain curves for HSCD of Fe-38Mn alloy are shown in Figure 2. As can be seen, the equivalent stress increases with the decrease in deformation temperature and the increase in strain rate. Moreover, in the range of selected deformation conditions, the flow curves show a continuous upward trend, and there is no obvious peak stress or steady-state flow image. Generally, for cylindrical compression, the flow stress rapidly increases due to the multiplication of dramatically developed dislocation, and work hardening dominates the deformation process at the beginning of deformation [29,30]. In addition, for low-stacking-fault-energy (SFE) austenitic steel, the flow curve usually has a peak value due to recrystallization during the thermal deformation process, and the deformation eventually reaches a steady-state flow when striking a balance between work hardening and dynamic softening [31]. The flow curve in this paper shows that the flow behavior of Fe-38Mn alloy during HSCD has the characteristics of continuous work hardening. Given the particularity of the HSCD and special alloy system, the influence of deformation conditions and microstructural evolution was considered comprehensively.

### 3.2. Microstructural Characteristics

Typical macroscopic deformation microstructural characteristics of HSCD are shown in Figure 3a–c, where the red arrow in Figure 3a represents the shearing direction. As shown, the deformation is mainly concentrated in the gage region of the SCS, which constitutes uniform shear-compression deformation with no significant strain localization. However, it can be seen from the overall deformation zone that the gage region narrows and the shear deformation becomes more obvious with the increase in strain rate; that is to say, the effect of shear deformation in HSCD gradually increases with the increase in strain rate. Furthermore, from the magnified microstructure in EBSD grain images shown in Figure 3a_1_–c_1_, it can be surmised that the DRX temperature and the grain size decrease with the increase in the strain rate. In addition, for further analysis of the DRX, the grain orientation spread (GOS) method was used to analyze the fraction of recrystallized grains; the EBSD GOS maps are shown in Figure 3a_2_–c_2_. In this method, a GOS value < 2° is defined as a DRX grain (the blue area in the GOS maps); otherwise, it is a deformed grain (the yellow and green area in the GOS maps) [32]. It was found that the proportion of the DRX increased from 15.4% at 900 °C, 0.01 s^−1^ to 44.5% at 900 °C, 10 s^−1^. Based on the observations of HSCD, it was noted that obvious DRX occurred in the microstructure, but there was no obvious peak in the curve. So, what exactly causes the continuous work hardening of Fe-38Mn alloy during HSCD? This issue will be addressed in the following discussion.

## 4. Discussion

### 4.1. Influence of Dynamic Microstructural Evolution

As mentioned above and the authors’ previous research [24], although obvious DRX occurred in the microstructure, the DRX of Fe-38Mn alloy during HSCD was particular. Deformation microstructures in the conditions of 900 °C and 0.01 s^−1^ are shown in Figure 4. In these, the low-angle grain boundaries (LAGBs, 2°~15°), high-angle grain boundaries (HAGBs, >15°) and twin boundaries (TBs) are highlighted by green, black and red lines, respectively, in Figure 4a, and the LAGBs, HAGBs and sub-grain boundaries (SGBs, 5°~15°) are highlighted by white, gray and black lines, respectively, in Figure 4b. With a low temperature and low strain rate, the DRX was characterized by the necklace feature (as shown in Figure 4a), which was considered conventional discontinuous DRX [33,34,35,36]. However, in the enlarged deformation area (as shown in Figure 4b) without obvious DRX, apart from the nucleation (shown by red dotted arrows) produced by a “bulged” mechanism, a large number of sub-grains (shown by the black arrows) are distributed along both sides of the serrated grain boundary with a clear orientation gradient, i.e., geometrically necessary grain boundary features. At the same time, there are also sub-grains inside the deformed grains, and even fine grains composed of HAGBs (shown by the red arrows). In addition, the TEM images in Figure 4c, d reveal that a large number of sub-grains were produced near the initial grain boundaries and inside the deformed grains, and the sub-grain boundary orientation was increased by consuming high-density dislocations, resulting in the subdivision of the original grains. Therefore, the dynamic microstructural evolution is a continuous DRX (cDRX) in nature pioneered by the “bulged” mechanism and accompanied by geometric characteristics and sub-grain migration characteristics.

It is noteworthy that dynamic microstructural evolution is closely related to shear banding in the process of HSCD. In order to further analyze the shear band problem in Fe-38Mn alloy, the parameter φ was introduced, as shown in Equation (1). Jonas [37] and Semiatin [38] proposed that based on flow curves, the parameter φ can predict strain localization and shear band problems in the thermal deformation of materials. Moreover, Jonas [36] pointed out that when φ > 5, the material has a tendency to display strain localization or shear bands.
(1)$$\varphi =\frac{1}{\dot{\varepsilon }}\frac{d\dot{\varepsilon }}{d\varepsilon }=-\frac{\gamma^{\prime }}{m}$$
where
(2)$$\gamma^{\prime }=\frac{1}{\sigma }\frac{\partial \sigma }{\partial \varepsilon }{|}_{\dot{\varepsilon }}$$
where *m* and *γ*′ are the strain rate sensitivity coefficient and work hardening/softening coefficient, respectively. Taking the two groups of parameters at 950 °C, 1 s^−1^ and 1100 °C, 10 s^−1^ as examples, the parameter φ at equivalent strains of 0.3, 0.6 and 0.9 was calculated, as shown in Table 1. It can be seen that the HSCD of Fe-38Mn alloy has a certain trend of strain localization or shear banding at relatively low temperatures, but it is not easy to produce at high temperatures, even at a high strain rate. No obvious strain localization or shear banding was found in the above flow curve and microstructural analysis, indicating that the shear localization in Fe-38Mn alloy has particularity.

Generally, the generation of macroscopic shear bands during deformation will lead to plastic instability or strain localization (for example, the macroscopic shear bands generated during HSCD of 7050 aluminum alloy [27]). In addition to dislocation movement, the generation of micro-shear bands within the grain provides another deformation mode, which is conducive to uniform deformation in the SPD process of low-SFE metals [39,40]. The HSCD has a combined deformation feature with a competition between compression and shear, and shear strain is the main strain path in the late stage of deformation [27]. Thus, in the conditions of 950 °C, 1 s^−1^, a large number of micro-shear bands with high-density entangled dislocations and dislocation cells are generated (as shown in Figure 5a), and it is notable that the bent/deformation twins are induced in micro-shear-band regions (as with the dotted circle shown in Figure 5b). Furthermore, and particularly, the interactions between twins and dislocations further promotes deformation during HSCD; the specific mechanism of the interactions is described in the authors’ previous articles [21] and will not be repeated here.

### 4.2. Work Hardening Behavior

In order to explore the characteristics of continuous work hardening, the work hardening rate *θ* (*θ = dσ/dε*) was analyzed, as shown in Figure 6. Figure 7 shows the microstructures of initial and typical stages with strain levels of 0.3, 0.6 and 0.9 at 110 °C, 10 s^−1^, the grain boundaries are named the same as in Figure 4. As can be seen, the curve of the work hardening rate fluctuates clearly, and the fluctuation degree weakens with the increase in deformation temperature and the decrease in strain rate. We classified the curve into five stages. These distinct stages of hardening for the low-SFE alloys have been labeled here as A, B, C, D and E. The first, stage A, is found to be very similar to traditional deformation work hardening behavior for the dynamic recovery regime (stage III), which has been reported previously [1] and will not be discussed in detail here. This recovery stage is followed by stage B, with a rising work hardening rate caused by the formation of micro-shear bands and the interaction of pre-existing annealing twins and dislocation. At a lower deformation temperature, the work hardening rate increases significantly due to the formation of micro-shear bands (Figure 5b). As shown in Figure 7a,a’, the initial microstructure is mainly composed of HAGBs and TBs, and the pre-existing TBs survive for the short insulation time before the deformation. Thus, in this stage, a large number of pre-existing TBs participate in deformation, meaning that in our investigation, dislocation slip was hindered and dislocation density increased due to the interaction between pre-existing TBs and dislocation at high strain rates (as shown in Figure 7b,b’), which further increased the work hardening rate. Subsequently, in stage C, the work hardening rate decreases due to the dynamic softening effect enhanced by the instantaneously triggered DRX process. With the increase in deformation temperature, in this study, dynamic softening was enhanced significantly with the increase in DRX degree, but the work hardening rate was still greater than zero because it was dominated by cDRX; in addition, a quantity of fine deformation twins was generated (as shown in Figure 7c,c’). Stage C is interrupted at high strains by stage D, and the work hardening rate increases due to the formation of deformation twins at a low temperature or high strain rate. Especially at a high strain rate, a large number of twin boundaries generated by the DRX process participate in deformation, resulting in high-density tangled and cell dislocation structures. Moreover, in this study, the deformation twinning was further enhanced (as shown in Figure 7d,d’), which added a dragging force to dislocation glide to increase the work hardening rate. At 1100 °C and 0.01 s^−1^, the work hardening rate is basically zero because the weak twinning effect at a low strain rate allows work hardening and dynamic softening to balance each other. In the last stage, E, the work hardening rate decreases due to the combined action of shear deformation and DRX.

## 5. Conclusions

In this study, a simple and reasonable physical model was used, which corresponds closely to the actual deformation process under the existing thermal simulation test conditions. In addition, this paper complements and improves the existing understanding of the fundamental and key problems in the actual deformation process of metals, especially the role of high-temperature deformation twins in the deformation of materials. Based on the above-described investigations, the following conclusions are drawn:1.The equivalent stress vs. equivalent strain curves for HSCD of Fe-38Mn alloy exhibit the characteristics of continuous work hardening. HSCD constitutes uniform shear-compression deformation with no significant strain localization, and it is mainly concentrated in the gage region of the SCS. However, the shear action in HSCD gradually increases with the increase in strain rate.2.The dynamic microstructural evolution is a cDRX in nature, pioneered by the “bulged” mechanism and accompanied by geometric characteristics and sub-grain migration characteristics. The combined deformation, especially shear deformation in the late stage of deformation, accelerates the formation of micro-shear bands, deformation twins and, particularly, interactions between twins and dislocations.3.During HSCD, the work hardening rate curve is classified into five stages. High-temperature deformation twinning increases the work hardening rate, and an increasing strain rate is beneficial to twinning behavior and promotes work hardening.

## Figures and Tables

**Figure 1 materials-17-03641-f001:**
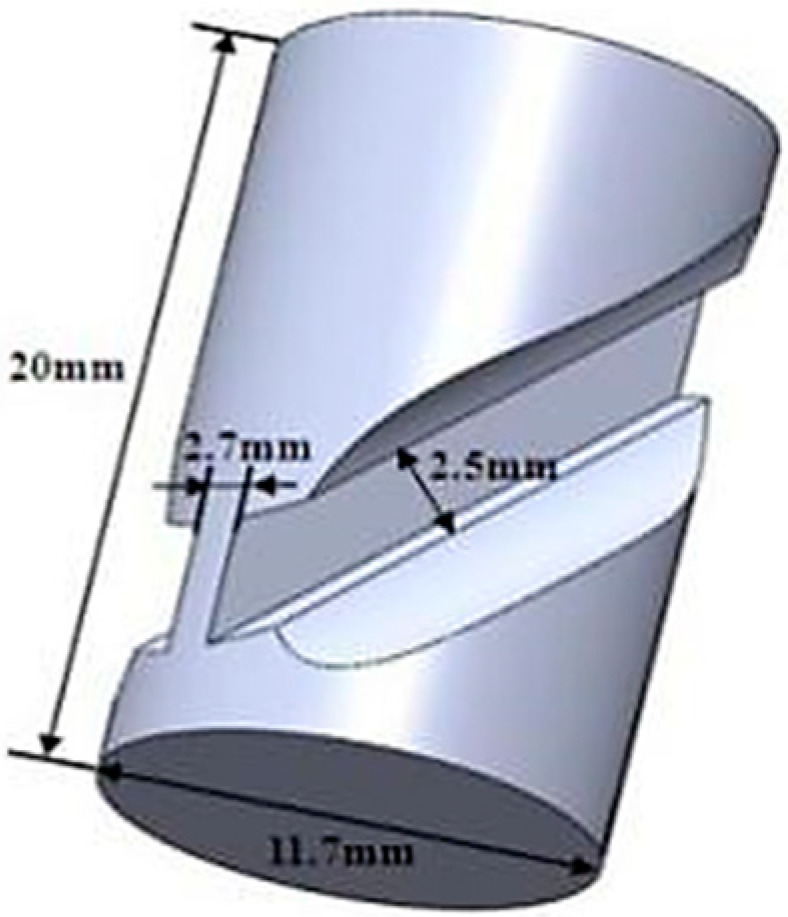
Schematic representation of SCS.

**Figure 2 materials-17-03641-f002:**
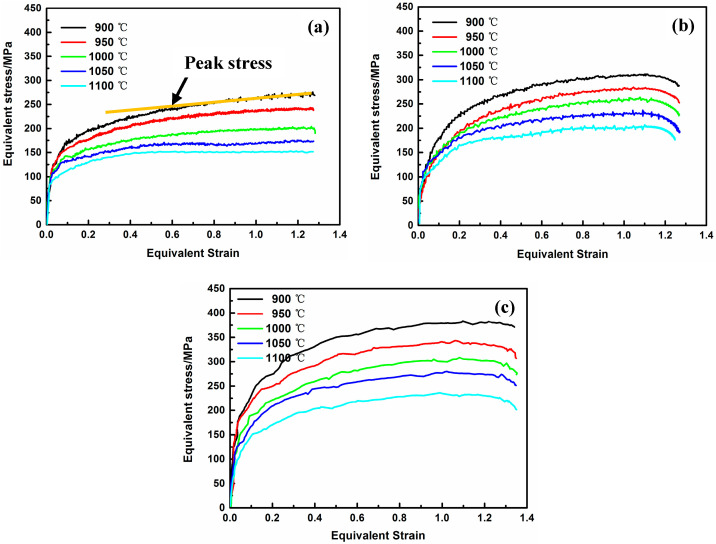
Equivalent stress vs. equivalent strain curves for HSCD of Fe-38Mn alloy at axial strain rates of (**a**) 0.01 s^−1^, (**b**) 1 s^−1^ and (**c**) 10 s^−1^.

**Figure 3 materials-17-03641-f003:**
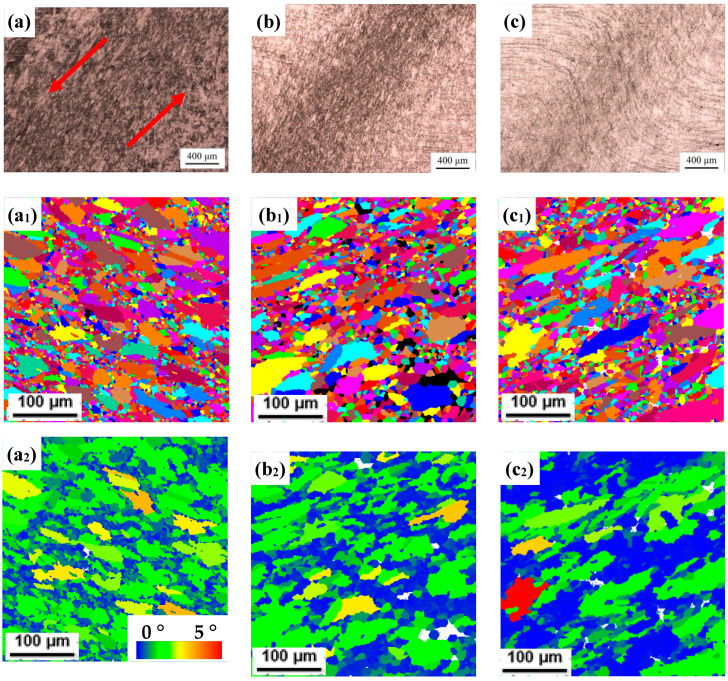
Typical deformation microstructures: OM images in the conditions of (**a**) 900 °C, 0.01 s^−1^, (**b**) 900 °C, 1 s^−1^ and (**c**) 900 °C, 10 s^−1^, EBSD Grain images in the conditions of (**a_1_**) 900 °C, 0.01 s^−1^, (**b_1_**) 950 °C, 1 s^−1^ and (**c_1_**) 900 °C, 10 s^−1^ and EBSD GOS maps in the conditions of (**a_2_**) 900 °C, 0.01 s^−1^, (**b_2_**) 950 °C, 1 s^−1^ and (**c_2_**) 900 °C, 10 s^−1^.

**Figure 4 materials-17-03641-f004:**
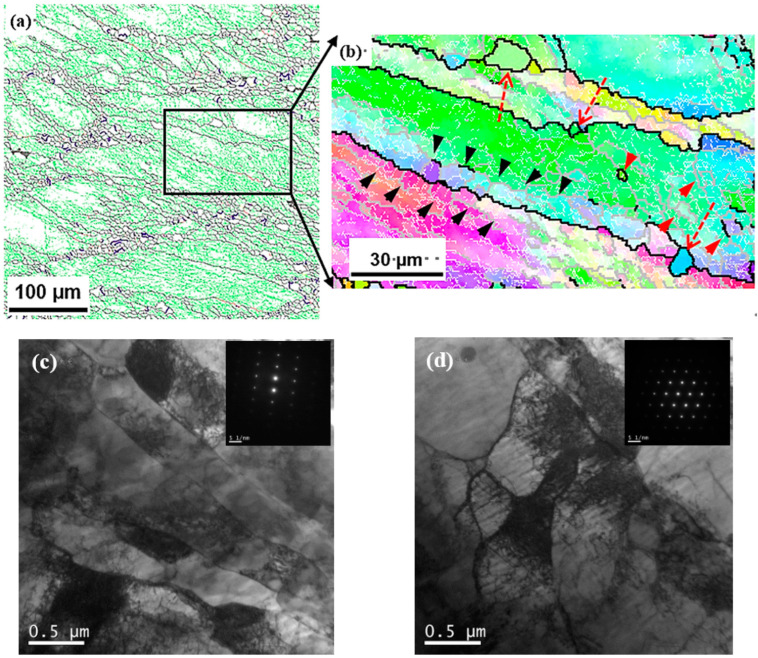
Deformation microstructures in the conditions of 900 °C, 0.01 s^−1^: (**a**) grain boundary distribution mappings, (**b**) OIM map of an enlarged region of (**a**), (**c**,**d**) TEM images.

**Figure 5 materials-17-03641-f005:**
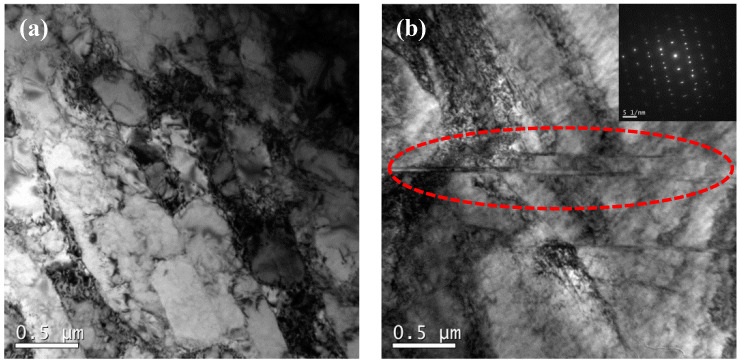
TEM images of the micro-shear bands (**a**) and bent/deformation twins (**b**) at 950 °C, 1 s^−1^.

**Figure 6 materials-17-03641-f006:**
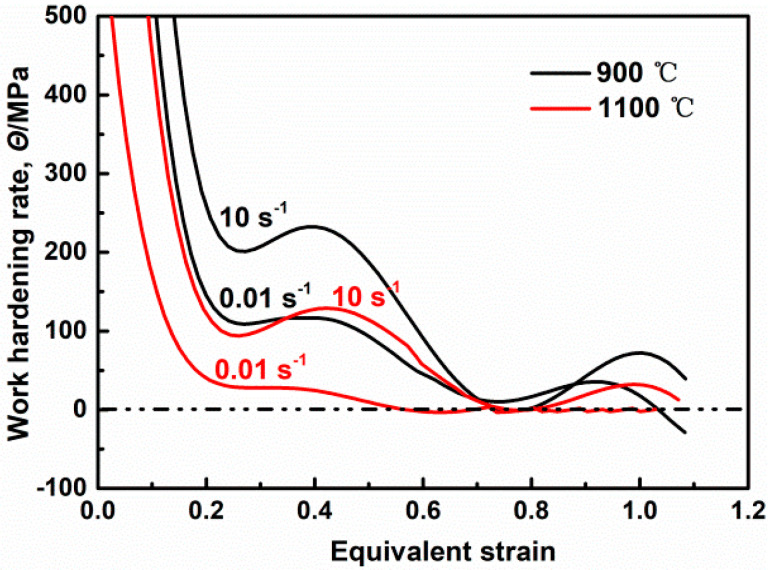
Work hardening rate vs. equivalent strain curves of Fe-38Mn alloy.

**Figure 7 materials-17-03641-f007:**
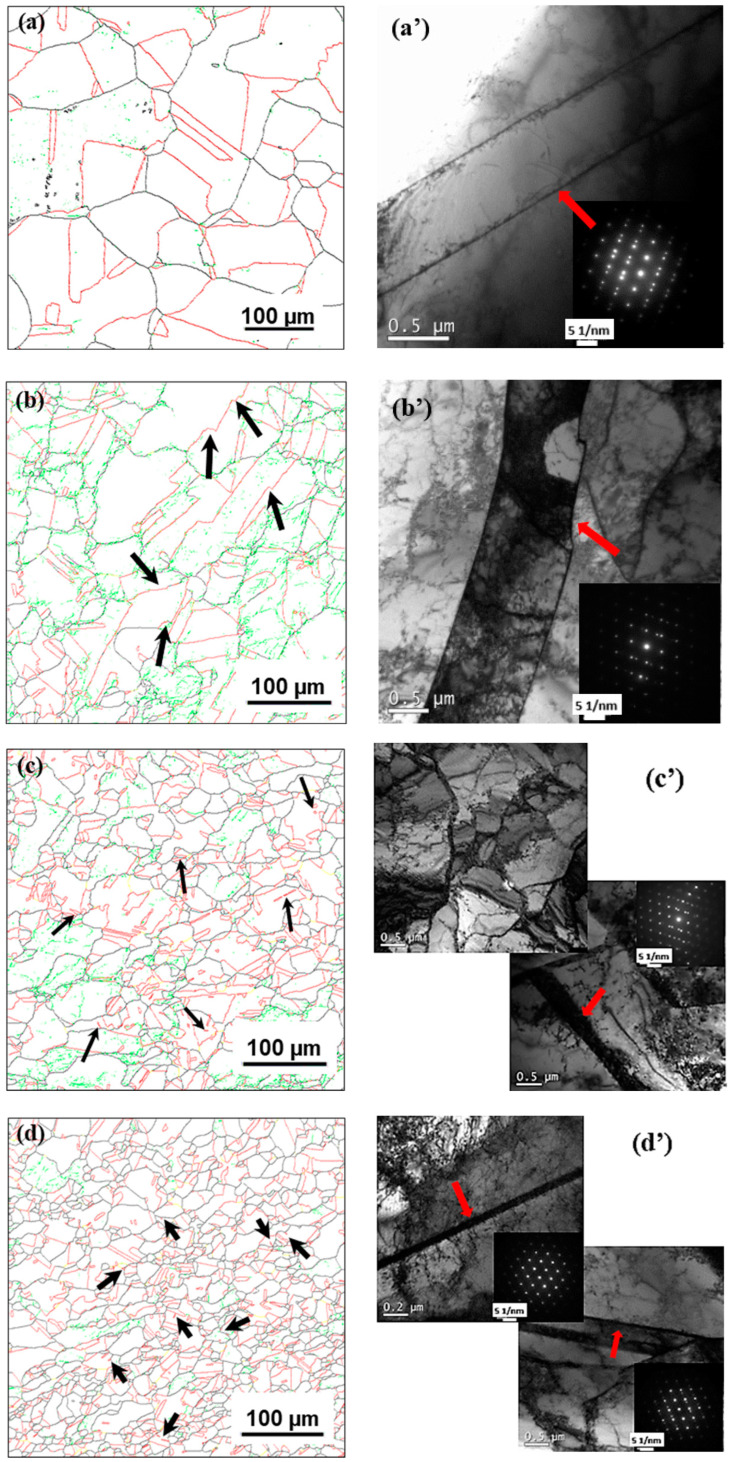
Microstructures of initial and typical stages with strain levels of 0.3, 0.6 and 0.9 at 110 °C, 10 s^−1^: (**a**–**d**) grain boundary distribution mapping of EBSD, (**a’**–**d’**) TEM images.

**Table 1 materials-17-03641-t001:** Values of the parameter *φ* in typical conditions with different equivalent strain levels.

Deformation Condition	Equivalent Strain		
	0.3	0.6	0.9
950 °C, 1 s^−1^	5.80	10.44	5.05
1100 °C, 10 s^−1^	<0	1.79	<0

## Data Availability

The original contributions presented in the study are included in the article, further inquiries can be directed to the corresponding author.

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
