# Peer review of "Effect of High-Temperature Deformation Twinning on the Work Hardening Behavior of Fe-38Mn Alloy during Hot Shear-Compression Deformation"

_materials, 2024, doi:10.3390/ma17153641_

Round 1

Reviewer 1 Report (New Reviewer)

Comments and Suggestions for Authors

Notes on the article of Deli Sang, Xiaoli Xin, Zikang Zhai, Ruidong Fu, Yijun Li and Lei Jing «Effect of High-Temperature Deformation Twinning on the Work Hardening Behavior of Fe-38Mn Alloy During Hot Shear-Compression Deformation»

The paper reports about the microstructure evolution and strengthening process of the Fe-38Mn alloy during hot shear-compression deformation. The authors showed that continuous work hardening occurs during hot shear-compression deformation. In this case, the work hardening rate curve has five stages. The evolution of the microstructure of the Fe-38Mn alloy during hot shear-compression deformation occurs along the path of dynamic recrystallization according to the “bulged” mechanism. The novelty of the work is questionable. However, the article can be published after revisions that are listed below, if the Editor makes this decision:

 1. More information regarding Fe-based alloys should be provided in the "Introduction" section. For example, the behavior High-Manganese Steel Fe-23Mn-0.6C was described in https://doi.org/10.1002/srin.201000263. Aspects of the recrystallization process in alloys with Mn content from 15 to 28% were also considered in https://doi.org/10.1016/j.msea.2022.144324. In general, out of 36 literature sources presented by the authors, only 4 are not older than 5 years. This should also be fixed.

2. The authors should state the novelty and purpose of the work in the “Introduction” section. Why were this material and this processing chosen? What is the reason for choosing the deformation temperature?

3. The authors described the “Experimental” section poorly. The process of melting of materials should be described in more detail and what row materials were used (especially what purity the materials were). Chemical analysis of the alloy (the actual composition) should also be provided. In addition, it is necessary to describe in detail the methodology for conducting microstructural studies: what equipment was used? How were the samples prepared? How were the samples etched (was the possibility of work hardening due to preliminary grinding and polishing of the samples taken into account)?

4. Figure 3. The authors should discuss in more detail the changes in structure at different strain rates and compare them quantitatively (by size or fraction of recrystallized grains).

5. What is the phase composition of the alloy after deformation?

6. The authors write: «Therefore, the dynamic microstructure evolution is a cDRX in nature that pioneered by “bulged” mechanism and accompanied by geometric characteristics and sub-grain migration characteristics». However, they did not previously provide a decoding of the abbreviation cDRX.

7. Figure 4 c and d. Electron diffraction patterns should be provided for TEM images.

Author Response

Comments 1:  More information regarding Fe-based alloys should be provided in the "Introduction" section. For example, the behavior High-Manganese Steel Fe-23Mn-0.6C was described in https://doi.org/10.1002/srin.201000263. Aspects of the recrystallization process in alloys with Mn content from 15 to 28% were also considered in https://doi.org/10.1016/j.msea.2022.144324. In general, out of 36 literature sources presented by the authors, only 4 are not older than 5 years. This should also be fixed.

Response 1: More information regarding Fe-based alloys have been provided in the "Introduction" section, and the references provided by the reviewers have been added. And added nearly 4 years of references.

Comments 2: The authors should state the novelty and purpose of the work in the “Introduction” section. Why were this material and this processing chosen? What is the reason for choosing the deformation temperature?

Response 2: The novelty and purpose of the work have been stated in the “Introduction” section. And the reason for choosing the material and processing have been added in page 2.

Comments 3: The authors described the “Experimental” section poorly. The process of melting of materials should be described in more detail and what row materials were used (especially what purity the materials were). Chemical analysis of the alloy (the actual composition) should also be provided. In addition, it is necessary to describe in detail the methodology for conducting microstructural studies: what equipment was used? How were the samples prepared? How were the samples etched (was the possibility of work hardening due to preliminary grinding and polishing of the samples taken into account)?

Response 3: The “Experimental” section have been described in more detail. Chemical analysis of the alloy and the methodology for conducting microstructural studies have been provided in page 2.

Comments 4: Figure 3. The authors should discuss in more detail the changes in structure at different strain rates and compare them quantitatively (by size or fraction of recrystallized grains).

Response 4: Figure 3. The changes in structure at different strain rates have discussed in more detail and compared them quantitatively by fraction of recrystallized grains in page 3-4.

Comments 5: What is the phase composition of the alloy after deformation?

Response 5: There is almost no change in phase composition after deformation, but there are very few new phases, which will be discussed in other articles.

Comments 6: The authors write: «Therefore, the dynamic microstructure evolution is a cDRX in nature that pioneered by “bulged” mechanism and accompanied by geometric characteristics and sub-grain migration characteristics». However, they did not previously provide a decoding of the abbreviation cDRX.

Response 6: The decoding of the abbreviation cDRX have been provided in page 4.

Comments 7: Figure 4 c and d. Electron diffraction patterns should be provided for TEM images.

Response 7: Figure 4 c and d. Electron diffraction patterns has been provided for TEM images in page 5.

Reviewer 2 Report (New Reviewer)

Comments and Suggestions for Authors

In the submitted paper, the authors investigated the effect of high-temperature deformation twinning on the work hardening behaviors of the Fe-38Mn alloy during hot shear-compression deformation. The authors demonstrated and proved that the flow curves for hot shear-compression deformation of the Fe-38Mn alloy exhibit the characteristics of continuous work hardening, which was uniform shear-compression deformation with no significant strain localization, but the shear action gradually increased with the increase of strain rate.

During the work on the manuscript, the authors observed that during hot shear-compression, the dynamic microstructure evolution was initiated by the "bulged" mechanism and had geometric and sub-grain migration characteristics, which was continuous dynamic recrystallization with deformation twinning. Based on the conducted research, the authors determined that due to the interaction between twins and dislocations or twins in deformation twinning, the work hardening rate was classified into five stages during hot shear-compression deformation.

The manuscript presented for review can be applied to solving practical engineering problems concerning the use of Fe-38Mn alloy, and the obtained results can be helpful during the production of semi-finished products under conditions similar to those of the study.

If I were the authors, I would slightly shorten and rephrase the abstract. I would not include results and conclusions but provide information that encourages the reader to go through the entire paper.

Even though the paper does not have many physical quantities, I would include a nomenclature section. There are many different symbols and abbreviations that require explanation. I ask the authors to add a nomenclature section, even at the end of the paper – it should be a complete list of abbreviations, symbols, and notations.

The introduction is a concise yet informative literature review.

I do not fully understand Figure 1. Some dimensions of the specimen are missing. Please replace this figure with a complete technical drawing, including all dimensions. You could also add a 3D model drawing of the specimen used in the studies.

Regarding the planning of the tests: Please provide detailed information on how many specimens the authors examined for each deformation level and temperature. To statistically compare the test results, there should be at least three specimens for each set of conditions. This allows for a full statistical analysis – minimum, maximum, median, mean, range, standard deviation, and sample size. From the graphic in Figure 2, it appears that only one specimen was tested at a given temperature and deformation level. In my opinion, there should be at least three specimens for each set of conditions. The final graph should then be the envelope of these results. Please clarify and address this comment.

Furthermore, regarding the tests: Please specify which signals were recorded during the test, how they were used to determine the equivalent stress and equivalent strain, and what conversion formulas were used to obtain these from the recorded signals.

What can the authors say about measurement errors? What are the errors associated with the measurements conducted by the authors?

Can the obtained curves be described by a constitutive relation or mathematical formula, and are there any correlations between the constants in this formula and the test temperature and strain rate? This could significantly enhance the paper.

I propose introducing designations for equivalent stress and equivalent strain in the manuscript.

Units in figures and tables should be written in square brackets with a space after the symbol or name of the quantity.

I have no comments on the discussion or the part of the paper concerning the assessment of the microstructure.

The conclusions should be slightly expanded. At the end, the authors should indicate the possible applications of the results presented in the manuscript in solving real engineering problems and their application in the industry. Furthermore, they should also outline directions for future research.

The paper has potential, but it requires revisions, expansion, and commentary. Please revise the manuscript and resubmit it for review. I suggest a major revision.

Comments on the Quality of English Language

Minor editing of English language required.

Author Response

Nomenclature section has been added at the end of the paper.

The introduction has been revised.

The contents proposed by the reviewer about Figure 1, the tests, equivalent stress and equivalent strain, measurement errors, constitutive relation and so on have been explained in the paper and discussed in detail in other published articles of the author. It is not appropriate to elaborate in this paper, please refer to the literature published by the author and cited in this paper.

Units in figures and tables have been written in square brackets with a space after the symbol or name of the quantity.

This paper is aimed at basic science issues, so the abstract and conclusion are based on customary expressions in this field without modification. Please kindly understand.

Round 2

Reviewer 1 Report (New Reviewer)

Comments and Suggestions for Authors

The article in this form looks normal. However, questions still arise about the novelty. Was the article accepted for publication?

Sang, Deli and Xin, Xiaoli and fu, ruidong and Li, Yijun and Jing, Lei, The Effect of High-Temperature Deformation Twinning on the Work Hardening Behavior of Fe-38mn Alloy During Hot Shear-Compression Deformation. Available at SSRN: https://ssrn.com/abstract=4543671 or http://dx.doi.org/10.2139/ssrn.4543671

 Unfortunately, at the moment it still available and it has free access.

Author Response

Comments1: The article in this form looks normal. However, questions still arise about the novelty. Was the article accepted for publication?

Response 1: Still under revision, unpublished.

Reviewer 2 Report (New Reviewer)

Comments and Suggestions for Authors

The authors, in the resubmitted paper and in their response to the review, suggest that they have made the necessary corrections to the paper and even added a nomenclature section. However, the paper submitted in the MDPI system by the authors is almost identical to the original version. I maintain my previous review and ask for the appropriate corrections to be made to the paper. For convenience, I am including the review once again.

In the submitted paper, the authors investigated the effect of high-temperature deformation twinning on the work hardening behaviors of the Fe-38Mn alloy during hot shear-compression deformation. The authors demonstrated and proved that the flow curves for hot shear-compression deformation of the Fe-38Mn alloy exhibit the characteristics of continuous work hardening, which was uniform shear-compression deformation with no significant strain localization, but the shear action gradually increased with the increase of strain rate.

During the work on the manuscript, the authors observed that during hot shear-compression, the dynamic microstructure evolution was initiated by the "bulged" mechanism and had geometric and sub-grain migration characteristics, which was continuous dynamic recrystallization with deformation twinning. Based on the conducted research, the authors determined that due to the interaction between twins and dislocations or twins in deformation twinning, the work hardening rate was classified into five stages during hot shear-compression deformation.

The manuscript presented for review can be applied to solving practical engineering problems concerning the use of Fe-38Mn alloy, and the obtained results can be helpful during the production of semi-finished products under conditions similar to those of the study.

If I were the authors, I would slightly shorten and rephrase the abstract. I would not include results and conclusions but provide information that encourages the reader to go through the entire paper.

Even though the paper does not have many physical quantities, I would include a nomenclature section. There are many different symbols and abbreviations that require explanation. I ask the authors to add a nomenclature section, even at the end of the paper – it should be a complete list of abbreviations, symbols, and notations.

The introduction is a concise yet informative literature review.

I do not fully understand Figure 1. Some dimensions of the specimen are missing. Please replace this figure with a complete technical drawing, including all dimensions. You could also add a 3D model drawing of the specimen used in the studies.

Regarding the planning of the tests: Please provide detailed information on how many specimens the authors examined for each deformation level and temperature. To statistically compare the test results, there should be at least three specimens for each set of conditions. This allows for a full statistical analysis – minimum, maximum, median, mean, range, standard deviation, and sample size. From the graphic in Figure 2, it appears that only one specimen was tested at a given temperature and deformation level. In my opinion, there should be at least three specimens for each set of conditions. The final graph should then be the envelope of these results. Please clarify and address this comment.

Furthermore, regarding the tests: Please specify which signals were recorded during the test, how they were used to determine the equivalent stress and equivalent strain, and what conversion formulas were used to obtain these from the recorded signals.

What can the authors say about measurement errors? What are the errors associated with the measurements conducted by the authors?

Can the obtained curves be described by a constitutive relation or mathematical formula, and are there any correlations between the constants in this formula and the test temperature and strain rate? This could significantly enhance the paper.

I propose introducing designations for equivalent stress and equivalent strain in the manuscript.

Units in figures and tables should be written in square brackets with a space after the symbol or name of the quantity.

I have no comments on the discussion or the part of the paper concerning the assessment of the microstructure.

The conclusions should be slightly expanded. At the end, the authors should indicate the possible applications of the results presented in the manuscript in solving real engineering problems and their application in the industry. Furthermore, they should also outline directions for future research.

The paper has potential, but it requires revisions, expansion, and commentary. Please revise the manuscript and resubmit it for review. I suggest a major revision.

Comments on the Quality of English Language

Minor editing of English language required.

Author Response

Comments1: In the submitted paper, the authors investigated the effect of high-temperature deformation twinning on the work hardening behaviors of the Fe-38Mn alloy during hot shear-compression deformation. The authors demonstrated and proved that the flow curves for hot shear-compression deformation of the Fe-38Mn alloy exhibit the characteristics of continuous work hardening, which was uniform shear-compression deformation with no significant strain localization, but the shear action gradually increased with the increase of strain rate.

Response 1: It is judged according to the deformation width and shear degree of the metallographic structure in Figure 3 (a-c).

Comments2: During the work on the manuscript, the authors observed that during hot shear-compression, the dynamic microstructure evolution was initiated by the "bulged" mechanism and had geometric and sub-grain migration characteristics, which was continuous dynamic recrystallization with deformation twinning. Based on the conducted research, the authors determined that due to the interaction between twins and dislocations or twins in deformation twinning, the work hardening rate was classified into five stages during hot shear-compression deformation.

Response 2. The "bulged" mechanism only starts at the initial stage of deformation, and the dominant position is still continuous recrystallization. For detailed discussion, refer to the published content of the author, which has been marked in the paper.

Comments3: The manuscript presented for review can be applied to solving practical engineering problems concerning the use of Fe-38Mn alloy, and the obtained results can be helpful during the production of semi-finished products under conditions similar to those of the study. Comments15: The conclusions should be slightly expanded. At the end, the authors should indicate the possible applications of the results presented in the manuscript in solving real engineering problems and their application in the industry. Furthermore, they should also outline directions for future research.

Response 3. 15. The conclusions have been slightly expanded.

Comments4: If I were the authors, I would slightly shorten and rephrase the abstract. I would not include results and conclusions but provide information that encourages the reader to go through the entire paper.

Response 4. The abstract has been slightly shortened and rephrased.

Comments5: Even though the paper does not have many physical quantities, I would include a nomenclature section. There are many different symbols and abbreviations that require explanation. I ask the authors to add a nomenclature section, even at the end of the paper – it should be a complete list of abbreviations, symbols, and notations.

Response 5. Nomenclature section has been added at the end of the paper.

Comments6: The introduction is a concise yet informative literature review.

Response 6. Thank you.

Comments7: I do not fully understand Figure 1. Some dimensions of the specimen are missing. Please replace this figure with a complete technical drawing, including all dimensions. You could also add a 3D model drawing of the specimen used in the studies.

Response 7. Figure 1 with a complete technical drawing has been replaced, and the 3D model drawing of the specimen has been added in page 2.

Comments8: Regarding the planning of the tests: Please provide detailed information on how many specimens the authors examined for each deformation level and temperature. To statistically compare the test results, there should be at least three specimens for each set of conditions. This allows for a full statistical analysis – minimum, maximum, median, mean, range, standard deviation, and sample size. From the graphic in Figure 2, it appears that only one specimen was tested at a given temperature and deformation level. In my opinion, there should be at least three specimens for each set of conditions. The final graph should then be the envelope of these results. Please clarify and address this comment.

Response 8. Three samples are tested under each parameter, and the one with the best deformation is taken as a reference. This explanation has been added in the revised manuscript. It is customary not to mark the deviation on the thermal deformation curve. The introduction has been revised.

Comments9: Furthermore, regarding the tests: Please specify which signals were recorded during the test, how they were used to determine the equivalent stress and equivalent strain, and what conversion formulas were used to obtain these from the recorded signals.

Response 9. The conversion relationship between equivalent strain-equivalent stress curve and axial displacement-loading force has been described in detail in many literatures, and reference sources have been added in this paper, so it will not be detailed here.

Comments10: What can the authors say about measurement errors? What are the errors associated with the measurements conducted by the authors?

Response 10. The measurement error is related to the selected method, and the measurement error is not a parameter that must be provided under some conditions.

Comments11: Can the obtained curves be described by a constitutive relation or mathematical formula, and are there any correlations between the constants in this formula and the test temperature and strain rate? This could significantly enhance the paper.

Response 11. This paper does not focus on the construction of constitutive relations, and the correlation problem is the author's follow-up work. The main purpose of this paper is to arouse the research on the high-temperature deformation twin problem as soon as possible.

Comments12: I propose introducing designations for equivalent stress and equivalent strain in the manuscript.

Response 12. The designations for equivalent stress and equivalent strain has been described in detail in many literatures, and reference sources have been added in this paper, so it will not be detailed here. If reviewer is interested in the specific construction parameters of this material, it is recommended to refer to other published articles of the author and cited in this paper.

Comments13: Units in figures and tables should be written in square brackets with a space after the symbol or name of the quantity.

Response 13. Units in figures and tables have been written in square brackets with a space after the symbol or name of the quantity.

Comments14: I have no comments on the discussion or the part of the paper concerning the assessment of the microstructure.

Response 14. Thank you.

Comments16: The paper has potential, but it requires revisions, expansion, and commentary. Please revise the manuscript and resubmit it for review. I suggest a major revision.

Response 16. The manuscript has been revised as requested, thank you.

This manuscript is a resubmission of an earlier submission. The following is a list of the peer review reports and author responses from that submission.

Round 1

Reviewer 1 Report

Comments and Suggestions for Authors

The manuscript by Xin et al. describes the effect of high-temperature deformation on the microstructure and mechanical properties of Fe-38Mn during friction stir processing. It is a reasonable kind of paper to be included in the special issue being planned on Advanced Forming Technologies. However, the paper is not suitable for publication in its present form for reasons given below.

Although the authors claim that the friction stir processing they have done causes the tensile strength and elongation to be greatly improved, that statement is not borne out by the presented results. The tensile stress-strain curves show clearly that the processing hardly changes the mechanical properties at all. There is some hint in the microhardness data that there could be some effect of friction stir processing, but those results may be compromised by some sort of contamination by tungsten-rhenium. The absence of significant effects on mechanical properties is not surprising. Although these is some grain refinement in processing zone or stir zone, the grains there are still quite large and not much smaller than those in the BM, so that little strengthening, especially for an already strong alloy like Fe-38Mn, should be expected. Also, for the tensile stress-strain curves the resulting properties are influenced both by the friction stir processing and by the unprocessed base metal. So, again small effects are not unexpected.

The wording in various places in the manuscript is hard to understand and should be improved:

In the first paragraph the following words make no sense:… “used in the materials industry to fine microstructure refinement in order to obtain fine recrystallization structure.”

On the first page: “..the thermal deformation of low-layer fault-energy FCC metals,” What does thermal deformation mean? What does low-layer fault-energy mean?

Also: “.. low-layer fault-energy crystal structures.” Does this mean low SFE crystal structures?

Also: “..for high-rise discharged aluminum alloys.” What does high-rise mean?

Also: “and monophasic austenite system.” Does this mean single phase?

Fig 1, It is not clear which XRD pattern is which?

Fig 5 better placement of boundaries between BM and PZ is needed to understand these results

The authors mention “tungsten-rhenium alloy residue of the mixing head.” That needs to be explained. It sounds like contamination related to the previous processing of tungsten-rhenium alloys.

The authors write: “From the curves of BM and engineering stress-engineering strain (as shown in Figure 5a). They must mean Figure 6a.

Finally, the authors should explain how the present paper differs from their ref 11 which some of them already published.

Comments on the Quality of English Language

See review

Author Response

Dear Reviewer,

Thank you for the comments concerning our manuscript entitled “Effect of high-temperature deformation twinning on the mechanical properties of Fe-38Mn alloy during friction stir processing”. Those comments are all valuable and very helpful for revising and improving our paper. We have studied comments carefully and have made correction which we hope meet with approval. We tried our best to improve the manuscript and made some changes in the manuscript. These changes will not influence the content and framework of the paper.

Reviewer 2 Report

Comments and Suggestions for Authors

In general, the content of this paper is not well organized. The logic and flow of the story need to be improved. The content in results and discussion sections are mixed: some text in results should be moved to discussion, while Fig. 5 to Fig. 7 and relavent desciptions should be moved to results. There are several additional comments as follows:

1. Fig. 1: lack of legend and detailed description, e.g. which curve is BM, which curve is PZ?

2. Please clarify what abnormal structues were observed in Fig. 2.

3. Fig 3: please add legend. In addition, the colors of different types of grain boundaries are not clear in the map, especially yellow line.

4. Please clarify what type of TEM images for Fig. 2 and Fig. 4, such as TEM-bright field image.

5. Fig. 5: please point out each zone in hardness curve.

6. Fig. 7: please clarify what type of SEM images, such as secondary electron micrograph.

Comments on the Quality of English Language

Overall, English can be improved.

Author Response

(The authors gave the same response as above.)

Reviewer 3 Report

Comments and Suggestions for Authors

Manuscript titled "Effect of High-Temperature Deformation Twinning on the Mechanical Properties of Fe-38Mn Alloy during Friction Stir Processing" is a nice work related to FSP of Fe-38Mn alloys.

Recommendation: Review Again After Resubmission (Paper is not acceptable in its current form, but has merit. A major rewrite is required. Author should be encouraged to resubmit a rewritten version after the changes suggested in the Comments section have been completed.)

Before publishing, the authors must revise the paper as per comments mentioned below:

1.     Recrystallized grains and fine deformed grains were observed in the structure of the stirred zone. The proportion of twin boundary is as high as 35% in the fine deformed grains. Why?

2.     How does pre-storage twins help in nucleation rate of dynamic recrystallization during the tensile process?

3.     Last paragraph of Introduction Section should highlight research gaps, and main objectives of this work

4.     Grain boundary results of Fig. 3 a and b should be compared in detail

5.     Microstructure results of Fig. 4 a, b and c should be discussed and compared well in more details

6.     The paper can be accepted after these corrections

Comments on the Quality of English Language

Minor spell check required

Author Response

(The authors gave the same response as above.)

Round 2

Reviewer 2 Report

Comments and Suggestions for Authors

The revised version is slightly improved compared with the first version. However, it still has more space to be improved in general.

Comments on the Quality of English Language

Can be improved.
